# A Review of the Emerging Poultry Visceral Gout Disease Linked to Avian Astrovirus Infection

**DOI:** 10.3390/ijms231810429

**Published:** 2022-09-09

**Authors:** Linlin Li, Minhua Sun, Yun Zhang, Ming Liao

**Affiliations:** 1Institute of Animal Health, Guangdong Academy of Agricultural Sciences, Guangzhou 510640, China; 2Key Laboratory for Prevention and Control of Avian Influenza and Other Major Poultry Diseases, Ministry of Agriculture and Rural Affairs, Guangzhou 510640, China; 3Key Laboratory of Livestock Disease Prevention and Treatment of Guangdong Province, Guangzhou 510640, China; 4The Brain Cognition and Brain Disease Institute (BCBDI), Shenzhen Institute of Advanced Technology, Chinese Academy of Sciences, Shenzhen 518055, China; 5Guangdong Laboratory for Lingnan Modern Agriculture, Guangzhou 510642, China; 6Guangdong Academy of Agricultural Sciences, Guangzhou 510640, China

**Keywords:** avian astrovirus, pathogenesis, visceral gout

## Abstract

Avian astroviruses, including chicken astrovirus (CAstV), avian nephritisvirus (ANV), and goose astrovirus (GoAstV), are ubiquitous enteric RNA viruses associated with enteric disorders in avian species. Recent research has found that infection of these astroviruses usually cause visceral gout in chicken, duckling and gosling. However, the underlying mechanism remains unknown. In the current article, we review recent discoveries of genetic diversity and variation of these astroviruses, as well as pathogenesis after astrovirus infection. In addition, we discuss the relation between avian astrovirus infection and visceral gout in poultry. Our aim is to review recent discoveries about the prevention and control of the consequential visceral gout diseases in poultry, along with the attempt to reveal the possible producing process of visceral gout diseases in poultry.

## 1. Review of the Poultry Astrovirus Classification

Astroviruses, belonging to the Astroviridae family and the genus *Avastrovirus* [1], are classified into two genera, including *Mamastrovirus* and *Avastrovirus* [2]. Mamastroviruses infect a wide range of animal taxa, including humans, cows, sheep, pigs, dogs, cats and rats, causing mild gastroenteritis [3,4], whereas Avastroviruses, firstly reported in 1965 [5], mainly infects turkeys, chickens, ducks, gooses, and pigeons [6,7,8,9,10]. Moreover, chicken-origin astroviruses detected in ducks are consistent with the molecular characterization of those of chicken derived sequences [9].

Avastroviruses spread rapidly around the world, causing great economic losses to the poultry industry and posing a threat to human health [11,12]. The viruses are typically transmitted horizontally within flocks via the fecal–oral route; the transportation of breeding eggs in poultry breeding areas can also lead to the prevalence of astrovirus, causing nephritis in chickens, fatal hepatitis in ducklings, and visceral gout in broilers and gooses, depending on the species of the virus [13,14]. Like other RNA viruses, astrovirus polymerase has no proofreading function in the process of replication. It is easy to make mistakes in the process of virus replication, leading to an increase in the mutation rate of the virus, and high genetic diversity and recombination potential [1,4,12,15]. This recombination between different strains of astroviruses has aggravated the prevention of astrovirus infection [4,11,16].

According to the International Committee of Taxonomy of viruses (ICTV) [17], Avastroviruses contain three species, including Avastrovirus 1, Avastrovirus 2, and Avastrovirus 3 (Figure 1a). Avastrovirus 1 comprises turkey astrovirus1 (TAstV-1), which was firstly detected in turkey poults suffering from diarrhea in 1980 in the United Kingdom [18], and later was found in diarrheic poults in the 1980s in the United States [7]. Avastrovirus 2 comprises avian nephritisvirus (ANV), including two different serotypes: ANV1 and ANV2. The first ANV was isolated from 1-week-old broiler chickens in 1976 [8] and was characterized as the first astrovirus of chickens based on sequence similarity in 2000 [19]. Avastrovirus 3 comprises turkey astrovirus 2 (TAstV-2) [20] and duck astrovirus 1 (DAstV-1) [21,22,23]. Besides the above-mentioned three avastrovirus species listed in ICTV, other unassigned avastroviruses, such as chicken astrovirus (CAstV) [24], DAstV-2, DAstV-3, DAstV-4, and NpAstV, are also assigned to the genus *Avastrovirus* [1,14,23]. Moreover, a newly reported goose astrovirus (GoAstV) [14,25] that causes visceral gout in gosling and ducking [18,26] was unassigned and most closely related to viruses classified within species Avastrovirus 3. Two distinct phylogenetic clades of GoAstV were identified, named GoAstV-1 and GoAstV-2 [27,28,29]. The classification of these unassigned avastroviruses is very necessary to promote the research of avastroviruses.

Currently, there are no commercial drugs or vaccines available to treat and to prevent astroviruses in poultry. Therefore, good levels of biosecurity including accurate detection method, impeccable management system, strict quarantine and elimination will be efficient ways to control this virus. An extended interval between production cycles will help reduce the risk of the avian astrovirus infection [30]. Multiple age farming should be avoided to shed the virus, and poultry wastes should be collected and removed on a regular basis in such a way that passages within the poultry facilities are free of contamination. In addition, the prevention of renal injury is important for reducing the incidence of gout in goslings [31].

### 1.1. Genome Structure and the Capsid Protein of the Avian Astrovirus

Astroviruses are characterized as a star-shaped morphology with a size of approximately 28–30 nm in diameter [32]. They are non-enveloped, single-stranded, positive-sense RNA viruses with a genome of approximately 6.2–7.7 kb, containing a 5′-untranslated region (UTR), three open reading frames (ORF1a, ORF1b, and ORF2), a 3′-UTR and a poly(A) tail [1] (Figure 1b). ORF 1a and ORF 1b encode non-structural proteins including a serine protease, a viral genome-linked protein (VPg), and an RNA-dependent RNA polymerase (RdRp) (Figure 1b), which are required for genomic replication. ORF2 encodes capsid precursor proteins required for virion formation and is important to induce the host immune response [33]. Within the astrovirus viral genome, ORF2 was the most divergent, whereas ORF1b appeared to be the least divergent among the different ORFs [34]. In earlier work, the antigenic characterization of avian astroviruses has been limited by the absence of virus-specific antisera, and the detection and identification of CAstVs was dependent on ORF 1b sequences [35]. Later, researchers found that though the ORF 1b of CAstVs shares comparatively high identities on nucleotide sequence (>75%) and amino acid sequence (>80%), their capsid protein sequences still differ substantially [36].

As a surface protein of the viral particle, the astrovirus capsid protein is likely to be the major determinant of virus antigenicity, cell tropism, immune protection and altered pathogenicity [2,36,37]. Evidence from human astrovirus studies show that the N-terminal region (residues 1 to 415) of the capsid protein is located in the inside of the capsid, while the C-terminal region (residues 416 to end) encodes for the outer surface of the capsid including the star-like capsid spikes, thereby suggesting that the C-terminal region interactions with the host cell receptors, antigenic variation and the host’s immune response [38,39].

Sequence analysis of avian astrovirus capsid genes revealed extensive genetic variation and the presence of distinct genotypes of CAstV, ANV and TAstV-2 circulating in poultry [2]. Changes in capsid protein caused by mutations or recombination can affect the pathogenicity and antigenicity of the virus, which is of practical implications for virus detection, epidemiological studies and the development of potential vaccines against astrovirus infections [2]. Earlier studies on TAstV-2 and human astrovirus showed that isolates sharing as high as 85% capsid protein identity are still considered to be serotypically different [40].

### 1.2. Pathogenicity of the Avian Astrovirus

Avian astroviruses infect different avian species. CAstVs usually lead to three diseases: (1) runting stunting syndrome (RSS) and uneven flock performance, originally characterized by enteritis and poor weight gain in young broiler flocks [41]; (2) kidney disease with visceral gout causing high mortality in young broilers (up to 40%) [1], (3) “white chicks” hatchery disease or white chick syndrome (WCS) characterized by transient increase in mid to late embryo deaths and a reduction in hatchability [1,12,42]. Besides diarrhea, growth retardation, kidney and intestinal lesions, ANVs infection also causes gout, resulting in increased mortality [16,43]. GoAstVs cause growth repression, visceral urate deposition kidney swelling and gout [14,26]. Other astroviruses, such as TAstV, frequently lead to enteritis and diarrhea. TAstV-2 is shown to cause poult enteritis complex (PEC), poult enteritis mortality syndrome (PEMS), and poult enteritis syndrome (PES) [7,20]. In addition, DAstVs cause hepatitis with high morbidity and mortality in ducklings [44]. Although different astroviruses cause various diseases in poultry, some astroviruses may share similar syndromes [1,12]. For example, young chickens infected, respectively, by CAstV or ANV can both presented stunted growth, enteritis and kidney lesions [1,3,33]. In addition, CAstV infected chickens and GoAstV infected goslings, ducks and chickens can cause fatal visceral gout [1,14,26].

## 2. Visceral Gout Caused by the Avian Astroviruses

Astrovirus has wide tissue phagocytosis and diverse pathogenicity in addition to mainly causing intestinal diseases [12,45]. Several kinds of astrovirus infections lead to kidney diseases and visceral gout, especially in different avian species [1,12,16,26] (Figure 2).

### 2.1. CAstV

CAstV is usually characterized as an enteric virus that mainly infects the host through the fecal–oral route. It has also been implicated in the severe kidney disease of young broilers with visceral gout [12,16]. In addition, CAstV were also found in turkey flocks, as well as in ducks and pigeons [9,46].

Symptoms including visceral gout after CAstV infection were also reported worldwide [12,15]. Furthermore, CAstV-affected chicks showed distended ureters with uricacid deposition, and visceral and articular gout [16]. The major post-mortem findings were swollen kidneys, prominent ureters and visceral gout. CAstV-infected embryos revealed stunted growth, liver necrosis and enlarged kidney with deposition of urate crystals [15]. In 2011 and 2012, India reported a 40% mortality rate of broilers with visceral and articular gout [16], the pathogen of which was identified as India subgroup Biii CAstV [12,15]. When inoculation with CAstV-infected kidney homogenate, 1 day-old SPF chickens showed a high mortality (67.5–100%), and autopsy showed that all chickens suffered from swollen kidneys, prominent ureters, and visceral gout.

### 2.2. ANV

ANV-infected young chickens presented diarrhea, growth retardation, kidney damage, interstitialnephritis, tubulonephrosis, and uricosis (gout), resulting in an increased mortality [15,16,47]. ANV was detected in broiler chicken flocks with gout symptoms in India [15], while 80 gout specimens with a percentage of 38.75% detected positive either for CAstV or ANV, 27.5% or 8.75% of which were positive for CAstV or ANV alone, respectively [15]. The rate of Avian astrovirus (AAstV)-positive samples in China was 1.40%, and the positivity rates of CAstV and ANV were 0.66% and 0.41%, suggesting that CAstV and ANV circulate in chickens in China [48]. However, whether there is an ANV strain causing chicken gout in China is unknown.

In addition, ANV is also found to infect turkeys, ducks, geese and pigeons [9,46]. Investigations of the diarrhea fecal samples in a population of pigeons in Shanghai revealed a positive rate of 89% (40/45) of ANV, and one positive sample indicated a co-infection with both ANV and chicken astrovirus [6]. Phylogenetic analysis revealed that the pigeon viruses detected were evolutionarily closely related to chicken ANV [46], suggesting a wide host range of these viruses. Therefore, multiple astroviruses were the most frequently identified combination in both diseased and healthy flocks.

### 2.3. GoAstV

GoAstV was detected in 2016 in goslings presented symptom of visceral gout and enlarged leg joint with urate deposits [30,49,50]. The goslings infected with GoAstV grew slowly and had significantly reduced body weight from 5 dpi [51]. Other symptoms caused by GoAstV include hemorrhage, swellings of kidneys and visceral urate deposition [30], as well as white feces, and paralysis [27]. Outbreaks of GoAstV resulted in a mortality rate of 30% [26]. Uric acid disposition on the surface of the liver, kidney, heart and in the ureter and articular cavity were also observed in dead goslings [14,51,52]. Moreover, in 2020, Muscovy ducklings infected with GoAstV presented visceral gout with a mortality rate of up to 61% [53]. GoAstV-1 and GoAstV-2 viruses have been verified to be the causative agent of goose gout [10,27,29,54,55]. In addition, GoAstV could infect chickens under experimental conditions [14]. Chickens infected with the isolated GoAstV showed similar clinical symptoms to those observed in goslings, including shedding white feces and enlarged leg joints in chickens, The necropsy showed swollen kidneys and visceral gout [14].

## 3. Physiopathology of Viral Infection Induced Gout

### 3.1. Lesional Pattern of Visceral Gout after Avian Astroviruses Infection

Histopathological studies revealed that urate deposits in kidneys, necrosis and degeneration of the epithelial cells of the proximal convoluted tubules with infiltration of granulocyte, lymphocytic infiltrations, and interstitial nephritis, cyst-like formation in the crypts of cystic crypt of the duodenum in CAstV infected chickens [56]. Severe congestion in heart, necrotic foci and different degrees of degenerative changes and deposition of urate crystals around the central vein in liver, and mild to moderate interstitial nephrosis in kidneys were found in dead chicken embryos [15]. In the GoAstV-infected goslings, the renal tubular epithelial cells became degenerated and necrotic, the brush border structure of the proximal convoluted tubule was destroyed; proliferation of fibrous connective tissue and slight inflammatory cell infiltration in the renal interstitium, glycogen was deposited in glomerular and thicken mesangium [28,49]. In the GoAstV-infected ducklings, necrosis and degeneration of renal epithelial cells, accompanied by interstitial lymphocyte infiltration in the kidneys, hepatocyte arrangement disorder, swelling, steatosis and infiltration of inflammatory cells in the liver were showed in histopathology [26].

### 3.2. Gout Pathogenicity

In nature, lack of uricase leads to a high incidence of gout in humans and poultry [38], poultry and humans have similar purine nucleotide metabolic pathways, and they have something in common in uric acid syntheses and metabolism. Gout pain seriously affects human health and the development of the poultry industry. Currently, there is a lack of effective and low-side-effect medicine for the treatment of gouty pain in both humans and poultry [2,20,22]. There are many studies on drugs for human gout, but few on avian gout. It is reported that adenylyl cyclase subtype 1 (AC1) is involved in the regulation of gouty pain in chickens. A selective AC1 inhibitor NB001 produces an analgesic effect (not anti-inflammatory effect) on gouty pain and may be used for the future treatment of gouty pain in both humans and poultry [57].

Chicken is considered one of the ideal animal models for the study of gout. Gout is caused by the nucleation and growth of monosodium urate crystals in tissues in and around the joints, following long-standing hyperuricemia [58]. Hyperuricemia occurs as a result of excessive uric acid from hepatic metabolism or renal underexcretion [59], which is not only associated with the production of uric acid but also related to excretion from the kidneys. It is reported that reduced excretion of urate is the predominant cause of hyperuricemia in human gout [60]. Therefore, the factors causing kidney and urinary tract injury, or urine concentration and excretion disorder, can promote the formation of urate deposition. Excessive UA production and reduced excretion were also found in the pathogenesis of GoAstV-induced gout in goslings. Lesions on the liver and kidney, as well as increased expression or activity of UA-production-related enzymes, contribute to hyperuricemia and gout formation [58]. It is also reported that mammalian astroviruses could increase the permeability of epithelial cells [61]; thus, the increased permeability of kidney epithelial cells induced by astrovirus certainly might explain the development of gout [16]. If the formation rate of urate is greater than the excretion capacity of urinary organs, gout can be caused by urate deposition on the surface of viscera [62]. Visceral gout, which occurs mainly in birds, is a metabolic disease caused by impaired kidney function, followed by an accumulation of urate crystals in various organs [27].

### 3.3. Relationship between Gout and Avian Virus Infection

In avian species, different viruses such as IBV, ANV, CAstV and GoAstV have been reported to cause gout in different avian species [12,16,27], and a newly reported DAstV, designated as DAstV-5, which has been isolated in 5-day-old Beijing ducks, can lead to visceral urate in duckings [63]. It has been reported that within 894 kidney samples collected from gout-affected chicks, 373 (41.7%) samples were positive for CAstV alone, while 326 (36.4%) were positive for both CAstV and ANV, and 18 (2.0%) were positive for CAstV, ANV, and IBV [16,28,64]. GoAstV is believed to be the causal agent of kidney disease and visceral gout, experimental infections with GoAstV alone could not fully reproduce the disease as in the field [27], and other infectious agents together with GoAstV may collectively contribute to the development of gout in gosling [10,16,65]. The increasing severity of gout might be associated with coinfection of GoAstV and goose parvovirus (GPV) [65,66]. It is reported that collected tissue from gout-affected goslings in 12 goose farms were positive for both GPV and GoAstV by PCRs and the fluorescence mIHC staining, and both GoAstV and GPV antigens were detected in all the examined tissues [66]. Another study reported co-infection with GoAstV and GPV in goslings [65]. The clinical signs, gross and histologic lesions, and PCR testing confirmed GoAstV and GPV coinfection as the cause of death, gross lesions of tissues and organs matched lesions. Thus, the coinfection of GoAstV and GPV may provide a synergistic impact to exacerbate the severity of gout in goslings [65,66]. In addition, co-infection of GoAstV-1 with GoAstV-2 detected in gout goslings also suggests the importance of GoAstV infection in the mechanism of gout [55].

There are few reports on the mechanism of gout caused by these avian virus infections. Research on avian coronavirus found that a significant increase in xanthine oxidase (XOD) activity in serum and a significant increase in serum uric acid levels were related to nephropathogenic IBV infection with increased uric acid synthesis and inhibited peroxisome functions [67]. It has been reported that the NMDAR-AC1-TRPA1 pathway may participate in the regulation of gouty pain in chickens [57]. For gout in goslings, young goslings are high-risk groups of visceral gout. They not only lack urate oxidase, which oxidizes the insoluble uric acid (UA) to water-soluble allantoin [68], resulting in the elevation of blood UA, but also have immature kidney, which is prone to injury [28,69]. The susceptibility of gosling to visceral gout may be related to the immune activation in the gut-kidney axis during the first week of life [70,71]. In addition, geese are herbivorous, thus depending on dietary fiber to perform normal activities. High-protein diets could activate the TLR4/MyD88/NFκB pathway and induce both intestinal and renal inflammation in young goslings [68]. High-protein diets have been implicated in kidney injury, and gut microbiota dysbiosis associated with gout in goslings have also been demonstrated [71].

### 3.4. The Capsid Protein of Avian Astroviruses and Possible Relation with Gout

Features of the capsid protein of CAstV are considered to drive the pathogenesis into different syndromes. Based on the amino acid phylogenetic tree of CAstV capsid protein, a new amino acid sequences clade of CAstV capsid protein associated with specific broiler kidney disease with visceral gout, which might help to elucidate the association of specific strains with gout disease [12]. For instance, phylogenetic analysis of capsid protein sequence of 18 India CAstV isolates revealed a high identity of 92.0 to 99.2%, which clustered these strains into a group designated as Biii subgroup CAstV associated with gout [12,15,16]. Furthermore, three CAstV strains isolated in the Middle East associated with gout were also classified into the same Biii subgroup [12]. The high degree of capsid amino acid conservation of CAstV strains, isolated in India and the Middle East, both associated with severe kidney disease and visceral gout, supports the hypothesis that this CAstV strain is an etiological agent. The study of the capsid protein gene is further required to understand the mechanism of gout after virus infection.

A phylogenetic tree was constructed by comparing the amino acid sequence of capsid gene from 41 avian Astrovirus (Figure 3). The capsid protein of 41 avian Astrovirus was distributed in eight major clusters. Among them, the CAstV has been divided into two major antigen groups, Group A including Ai, Aii, and Aiii, and Group B including Bi, Bii, Biii, and Biv [12]. Capsid gene-based phylogenetic analysis revealed clustering of all the CAstV strains of gout origin into single group—the Biii CAstV antigenic group [15]. All GoAstV isolates associated with kidney disease and visceral gout were clustered into two subgroups, GoAstV-1 and GoAstV-2 (Figure 3). Though together associated with kidney disease and visceral gout, isolates in subgroup Biii and GoAstV share low amino acid sequence identity. In addition, western Canadian isolates together with CkP5/US/2016 and CC_CkAstV/US/2014, with clinical symptoms of white chick syndrome (WCS) [72], were clustered into a subgroup as Biv CAstV antigenic group.

In addition, the aa sequence identities of capsid gene of the 41 avian astrovirus within subgroups (Figure 3) were high: Ai—95.1–98.6%; Aii—98.6–98.9%; Aiii—98.0–99.5%; Bi—98.6–98.7%; Bii—94.2–97.7%; Biii—97.0–99.4%; Biv—97.0–99.4%, TAstV—81.8–97.5%, DAstV—94.3%, GoAstV-1—81.0–96.7%, and GoAstV-2—97.1–99.1%. Even though subgroup Biii CAstV and GoAstV associated with kidney disease and visceral gout, they only shared 34.3–35.1% of aa identity. The capsids protein of subgroups Biii CAstV comprised 738 amino acid, subgroup GoAstV-1 comprised 704–707 amino acids and subgroup GoAstV comprised 704 amino acids. Detailed examination of the subgroup Biii CAstV isolate’s capsid protein N-terminal region (residues 1 to 415) revealed it had 98.3 to 100.0% amino acid identity, while the C-terminal region (416 to the end) shared 95.4 to 100.0% identity. However, the amino acid identity of the capsid protein N-terminal region of the GoAstV isolate is very close to the C-terminal region. Though the N-terminal region of capsid protein was conserved, and C-terminal region was variable in subgroup Biii CAstV [72], there is a conserved C-terminal region between subgroups Biii CAstV and subgroup GoAstV-1 and GoAstV-2 (Figure 4a), which corresponding to residues 680–723 of subgroup Biii CAstV. A conserved region, corresponding to residues 319–361 of subgroup Biii CAstV was also found (Figure 4b). In addition, residues 128–132, 175–179, 345–349, 409–412 and 428–431 of subgroup Biii CAstV were conserved with subgroup GoAstV (Figure 4c). Because the C-terminal region codes for the outer surface of the capsid may interacting with the host cell receptors and antigenic variation, whether these conserved C-terminal region between subgroups Biii CAstV and subgroup GoAstV-1 and GoAstV-2are related to pathogenicity remains to be studied. There seems to be a relationship between visceral gout and specific groups of the avastroviruses, and future work is required to uncover the relationship between the capsid protein and visceral gout.

## 4. Perspectives

Belonging to the genus *Avastrovirus*, avastroviruses are endemic worldwide, causing significant loss to the poultry industry. Poultry flocks are susceptive to the virus, which infects a variety of avian species. Currently, there are no available drugs or vaccines to treat the virus, and high level of biosecurity is required to efficiently control the virus. Therefore, development in control measures of the virus is urgently required.

Gout is due to kidney damage with several factors [12,16,66]. Though the detection and isolation of astroviruses from gout cases suggest their possible role in disease, several factors including feed nutrition, poultry house management, and the presence of pathogenic microorganisms can cause gout in avian [12,28]. Further research is required to explore the role of these factors in gout.

Different avian viruses, including IBV, ANV, CAstV, etc., have been reported to cause gout. However, the mechanism underlying remains unclear. Since chicken serves as an ideal animal model to study pathogenesis of gout, CAstV- and GoAstV-infected chicken could be applied to analyze the mechanisms underlying. Phylogenetic analysis on gout related strains suggests a high similarity on amino acid sequences of the capsid proteins of these strains, suggesting the capsid protein play an important role not only in viral classification, but also in gout pathogenesis. Though with limited information according to the current research, future work exploring the function of the capsid protein would be interesting.

## Figures and Tables

**Figure 1 ijms-23-10429-f001:**
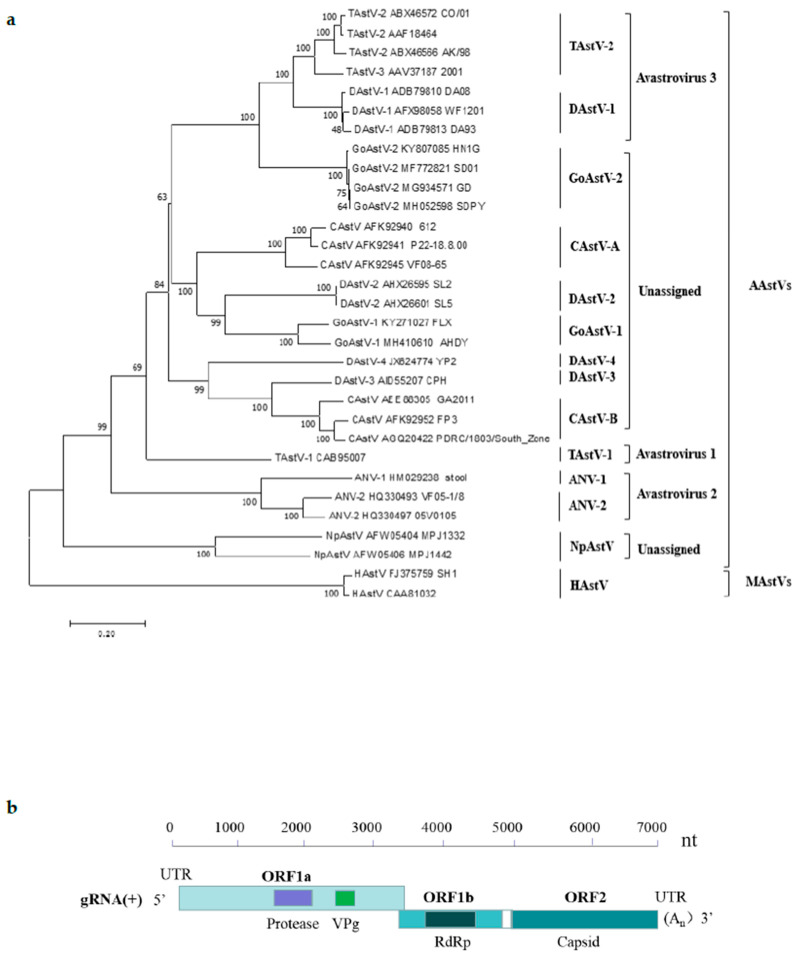
Phylogenetic tree of the amino acid sequences of astrovirus isolates based on the complete sequences of the capsid protein (**a**). All the reference sequences used in this study were obtained from the GenBank database. The tree was constructed by the neighbor-joining method with 1000 bootstrap replications using MEGA 7.0.14 software. Complete genome organizations of avian astroviruses (**b**). Nucleotide position of three ORFs in the genome is shown. ORF, open reading frame; VPg, viral genome-linked protein; RdRp, RNA- dependent RNA polymerase.

**Figure 2 ijms-23-10429-f002:**
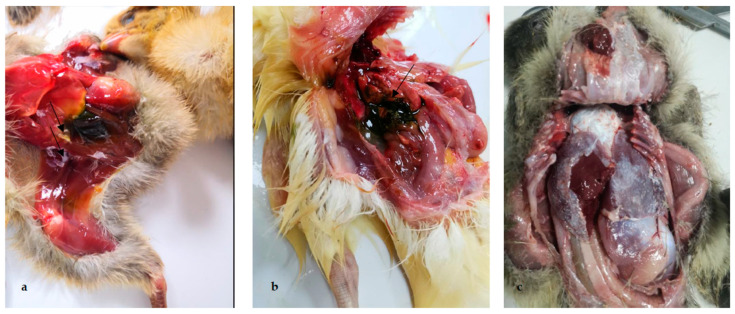
Pathological lesions of clinical samples of chickens, ducklings and goslings affected with gout. (**a**) Urate deposition on the peritoneum and between legs and abdomen of chicken; (**b**) urate deposition in the gallbladder of duckling; (**c**) urate deposits on pericardium and peritoneum of gosling.

**Figure 3 ijms-23-10429-f003:**
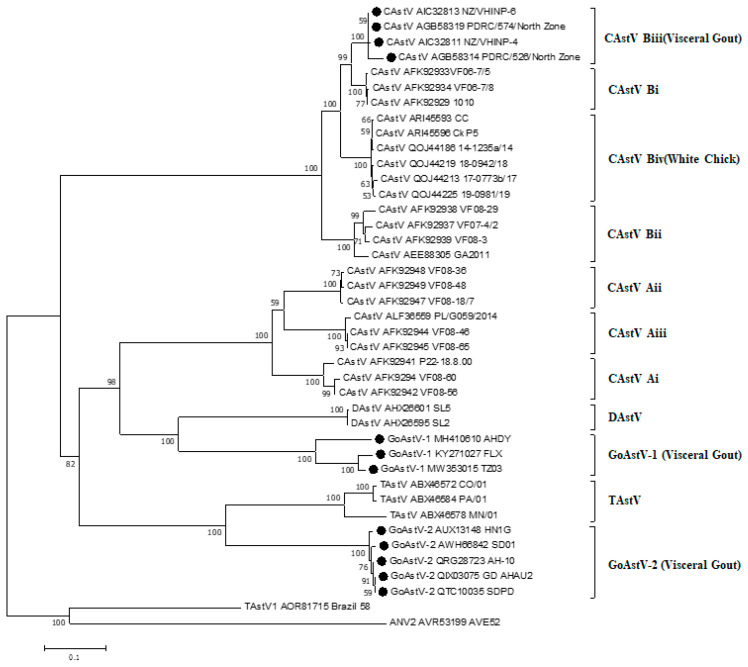
Phylogenetic tree of avian Astrovirus based on complete amino acid sequences of the capsid protein. The tree was constructed using Mega 7.0.14 software (DNASTAR Inc., Madison, WI, USA). using the neighbor-joining method and 1000 bootstrap replicates (bootstrap values are shown on the tree). Avian nephritis virus serotype 2 (ANV-2) was used to root the tree. Strains causing visceral gout were marked with black dots.

**Figure 4 ijms-23-10429-f004:**
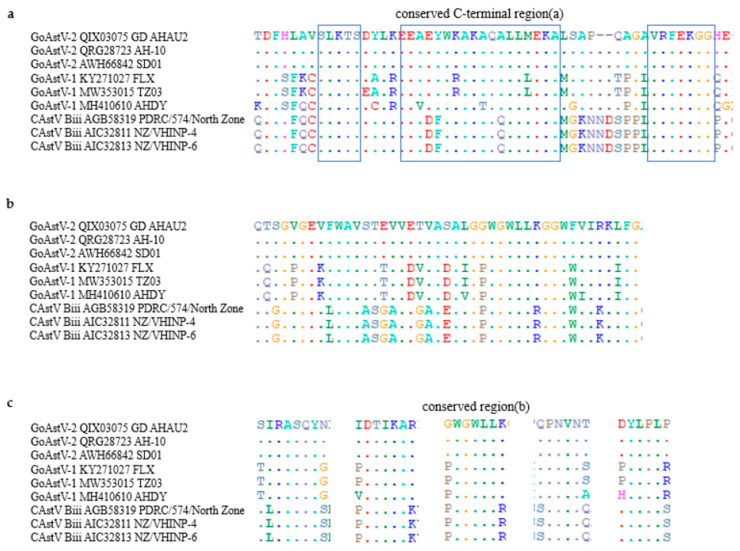
Clustal W alignment of the complete of the capsid proteins of nine representative isolates (subgroup Biii CastVs(3), subgroup GoAstV-1(3) and subgroup GoAstV-2(3)). The PDRC/574/North Zone, NZ/VHINP-4 and NZ/VHINP-6 belong to subgroup Biii, the FLX, TZ03 and AHDY belong to subgroup GoAstV-1, and the rest belong to subgroup GoAstV-2. The conserved C-terminal regions (residues 416 to end), corresponding to residues 680–723 of subgroup Biii CAstV (**a**). A conserved region, corresponding to residues 319–361 of subgroup Biii CAstV (**b**). An example of conserved region, corresponding to residues 128–132, 175–179, 345–349, 409–412 and 428–431 of subgroup Biii CAstV are shown (**c**).

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
