# Peer review of "A Review of the Emerging Poultry Visceral Gout Disease Linked to Avian Astrovirus Infection"

_ijms, 2022, doi:10.3390/ijms231810429_

Round 1
Reviewer 1 Report
Manuscript ID: ijms-1884927
Type of manuscript: Review
Title: A Review of the Emerging Poultry Visceral Gout disease after Avian Astrovirus Infection
Authors: Linlin LI, Minhua Sun, Yun Zhang *, Ming Liao * Submitted to section: Molecular Microbiology
2 Suggested changes in the title: ‘linked to’ instead of ‘after’
25 Suggestion to merge chapters 1,2 & 3: instead of a poorly informative title such as ‘introduction’ the three merged chapters could be named ‘Review of the poultry Astrovirus classification’ or something approaching considered by the authors as more relevant. The overall title of the paper refers to visceral gout
176 Suggestion to change the title: ‘Physiopathology’ instead of ‘Mechanisms’
177 Suggestion to substitute ‘Histopathology’ by ‘Lesional pattern’ of something similar more referring to the description, and not to the method used
193 Suggestion to be consistent with the former suggested substitution: ‘Gout pathogenicity’ or something similar according to the authors’ choice
Author Response
2 Suggested changes in the title: ‘linked to’ instead of ‘after’
Reply: Thank you for your valuable suggestions. The title has been changed to: “A Review of the Emerging Poultry Visceral Gout Disease Linked to Avian Astrovirus Infection”.
25 Suggestion to merge chapters 1,2 & 3: instead of a poorly informative title such as ‘introduction’ the three merged chapters could be named ‘Review of the poultry Astrovirus classification’ or something approaching considered by the authors as more relevant. The overall title of the paper refers to visceral gout
Reply: Thank you for your valuable suggestions. Chapters 1,2 & 3 has been merged, with a title “Review of the poultry Astrovirus classification”.
176 Suggestion to change the title: ‘Physiopathology’ instead of ‘Mechanisms’
Reply: Thank you for your valuable suggestions. The title has been changed to: “Physiopathology of Viral Infection Induced Gout” (line185).
177 Suggestion to substitute ‘Histopathology’ by ‘Lesional pattern’ of something similar more referring to the description, and not to the method used
Reply: Thank you for your valuable suggestions. The title has been changed to: “Lesional Pattern of Visceral Gout after Avian Astroviruses Infection” (line186).
193 Suggestion to be consistent with the former suggested substitution: ‘Gout pathogenicity’ or something similar according to the authors’ choice
Reply: Thank you for your valuable suggestions. The title has been changed to: “Pathogenicity of gout” (line202).
Reviewer 2 Report
Overall, the publication complies with the editorial requirements. However, there is no information about other astroviruses, e.g. in ducks and pigeons, where these viruses are important from an epidemiological point of view. It recommends supplementing the publication with the above data.
Author Response
Overall, the publication complies with the editorial requirements. However, there is no information about other astroviruses, e.g. in ducks and pigeons, where these viruses are important from an epidemiological point of view. It recommends supplementing the publication with the above data.
Reply: Thank you for your valuable suggestions. The epidemiological information on astroviruses in ducks and pigeons were added in the revised manuscript (Line 30- 32; Line 163-169).
